# HIGH-PERFORMANCE RNNS WITH SPIKING NEURONS

## ABSTRACT

The increasing need for compact and low-power computing solutions for machine learning applications has triggered significant interest in energy-efficient neuromorphic systems. However, most of these architectures rely on spiking neural networks, which typically perform poorly compared to their non-spiking counterparts in terms of accuracy. In this paper, we propose a new adaptive spiking neuron model that can be abstracted as a low-pass filter. This abstraction enables faster and better training of spiking networks using back-propagation, without simulating spikes. We show that this model dramatically improves the inference performance of a recurrent neural network and validate it with three complex spatio-temporal learning tasks: the temporal addition task, the temporal copying task, and a spoken-phrase recognition task. We estimate at least $500\times$ higher energy-efficiency using our models on compatible neuromorphic chips in comparison to Cortex-M4, a popular embedded microprocessor.

## 1 INTRODUCTION

Exponential growth in computational power and efficiency have played a vital role in the development of neural networks and their training algorithms. However, it has also led to higher design complexity and increasing difficulty to keep up with Moore's law (Schaller, 1997; Waldrop, 2016). Recent years have also seen the movement of computation from data-centres to compact, distributed, and portable embedded systems. These factors have created a demand for energy-efficient AI-capable devices, leading to the development of dedicated and optimized von Neumann-style Artificial Neural Network (ANN) accelerators (Aimar et al., 2018; Cavigelli and Benini, 2016; Chen et al., 2016) and a renewed interest in neuromorphic systems (Chicca et al., 2014; Frenkel et al., 2019; Davies et al., 2018; Moradi et al., 2018; Akopyan et al., 2015; Qiao et al., 2015; Neckar et al., 2019).

A key difference between neural networks deployed on von Neumann systems and most neuromorphic platforms is the use of spikes or train of pulses to represent signals in the latter. Such networks are called Spiking Neural Networks (SNNs). Spiking neuromorphic systems have a number of features that inhibit their use in real-world problems: (1) mixed-signal circuits suffer from Complementary Metal-Oxide-Semiconductor (CMOS) mismatch (Pelgrom et al., 1989) that degrades performance; (2) rate-based SNNs generate a large number of spikes to represent signals that reduces their energy benefit; (3) complex spiking dynamics makes it difficult to train them using gradient-descent methods. In this paper, we describe a new neuron model that addresses these problems and discuss how its Low-Pass Filter (LPF) abstraction enables training spiking Recurrent Neural Networks (RNNs) using the backpropagation algorithm (or Backprop). This is a significant breakthrough as it enables training and deployment of energy-efficient spiking neural network devices without simulating complex spiking dynamics.

## 2 PROCESSING-IN-MEMORY FOR RNNS

Consider an RNN layer with $n$ nodes. At each time-step, the processor computes one or several matrix products of the form $y = W.x$, where $y$ and $x$ are vectors of length $n$, and $W$ is a 2-D matrix of size $n \times n$. When operating on a von Neumann system with batch-size 1, as is common in most edge applications, the bottleneck in throughput and energy-efficiency is the $O(n^2)$ memory fetches of $W$ at every timestep. An in-memory matrix multiplier addresses this problem. It is a module where a "read" from the memory location of the $W$ matrix, using "X" as the "address" gives out "Y", without ever moving $W$. This leads to a quadratic reduction in energy consumption. Processing

in-memory systems have been implemented for various tasks such as DNA sequencing (Ghose et al., 2018), graph processing (Ahn et al., 2015), etc and with dramatic reduction in energy consumption.

The energy reduction from in-memory computing is well established, but the key challenge with deploying such systems is the absence of compatible algorithms. In this paper, we propose an RNN model for such a system. The implementation of the in-memory module depends on how $x$ and $y$ are encoded and transported. It can be synchronous or asynchronous and analogue or digital. We adopt an asynchronous digital approach as it offers some implementation advantages. Encoding information in binary digital format is less susceptible to noise in comparison to analogue. Asynchronous signalling allows the energy-consumption to scale in proportion to chip activity, while also permitting low-latency response. Chips implementing such schemes have been published in literature (Qiao et al., 2015; Moradi et al., 2018). The model presented in this paper is designed to integrate on similar chips (A reference framework is described in supplementary section D). However, most of the algorithmic ideas presented in this paper are general and applicable to a range of compute systems.

## 3 THE SPIKING NEURON MODEL

Spiking neuron models for encoding signals typically use rate- or time-coded spike-generation schemes (Diehl et al., 2015; Rueckauer et al., 2017; Bohte, 2012; Mostafa, 2017). In rate-coding, the firing rate of the neuron is proportional to the input signal. Therefore, achieving high data-resolution with rate-coding requires a large number of spikes, which is not energy-efficient (Nair and Indiveri, 2019). To address this problem, several time-coding schemes have been proposed. In this work, we build on existing models to propose an Adaptive Integrate and Fire (aI&F) neuron model, which can also be interpreted as an asynchronous $\Sigma\Delta$ circuit (Nair and Indiveri, 2019; Bohte, 2012; Yoon, 2016). This mechanism reduces the spike count by only transmitting the error between an internal state and the input. The aI&F neuron model implemented with current-mode neuromorphic circuits (Nair and Indiveri, 2019) can be described by the following equations:

$$\tau_{mem}\frac{dI_{mem}}{dt} = \alpha_L(I_L - I_{mem}) - s + i \tag{1}$$

$$\tau_w\frac{ds}{dt} = \alpha_s(I_{mem} - I_L) - s \tag{2}$$

$$I_{mem} = 0, \quad \text{when } I_{mem} > \Delta \tag{3}$$

where the currents $I_{mem}$ and $I_L$ represent the "membrane potential" and "leak reversal potential" variables. The term $s$ represents the neuron adaptation current, $i$ the input current, $\tau_{mem}$ the membrane time constant, $\alpha_L$ a gain factor, $\Delta$ the threshold, $\alpha_s$ the adaptation coupling parameter and $\tau_w$ is the adaptation time constant. The aI&F model is a feedback loop that tries to decrease the difference between the $i(t)$ and $s(t)$. The difference, $i(t) - s(t)$, is filtered with gain, $\alpha_L$, and time constant, $\tau_{mem}$. When the output of this filter, $I_{mem}$, exceeds the spiking threshold, $\Delta$, $I_{mem}$ is reset and a spike is generated. The $\Sigma\Delta$ circuit model used in this work is different from the aI&F model in the computation of the feedback term. Instead of filtering $I_{mem}$, we operate on the spike train generated by the spiking neuron. This ensures that the noise inserted by the spike-generation mechanism is also suppressed by the $\Sigma\Delta$ feedback. The modified feedback equation is as follows:

$$\tau_k\frac{ds}{dt} = \alpha_s(\delta_i I_{in} - I_L) - s \tag{4}$$

Where, $\delta_i$ indicates the spike train and $I_{in}$ is a programmable maximum current value that the analogue filter implementation can generate. The product $\delta_i I_{in}$ models the feedback filter that integrates a current $I_{in}$ for the duration of the spikes. Figure 1a shows a block diagram of the circuit implementation of the Equation 3 with the modification described by Equation 4. In this diagram, $F(s)$ is a first order LPF that receives inputs to the neuron. $H(s)$ is a first-order low-pass filter that produces $s(t)$ in response to spikes generated by the neuron. $E(s)$ is also a first-order low-pass filter on the difference between the input current $i(t)$ and the feedback signal $s(t)$. When the output of $E(s)$, $I_{mem}$, exceeds the spiking threshold ($\Delta$) of the neuron, a spike-event is produced. With each spike-event, $s(t)$ increases and $i(t) - s(t)$ decreases. Figure 1b shows the asynchronous $\Sigma\Delta$ feedback loop in action for a test-case. It can be shown that the Laplace domain representation of the output spike train can be expressed by the equation:

$$Y(s) = \frac{X(s)F(s)E(s)}{1 + H(s)E(s)} + \frac{N(s)}{1 + H(s)E(s)} \tag{5}$$

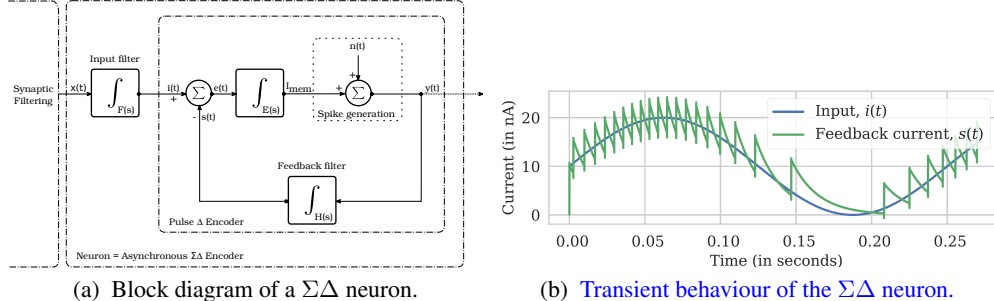

(a) Block diagram of a $\Sigma\Delta$ neuron.  (b) Transient behaviour of the $\Sigma\Delta$ neuron.

Figure 1: (a) The block marked pulse $\Delta$ encoder is similar to the aI&F neuron model. The LPF stage at the input makes it an asynchronous $\Sigma\Delta$ loop. (b) Evolution of the feedback signal, $s(t)$, over time. The sudden jumps in $s(t)$ correspond to spike events, $y(t)$.

where, $N(s)$ indicates the Laplace-domain representation of the noise introduced into the loop, for example by the spike generation mechanism. The $\Sigma\Delta$ loop described by Equation 5 is similar to a continuous-time $\Sigma\Delta$ modulation loop (Pavan et al., 2017) with the key difference being that the output spikes are unipolar. This is valuable because the output of a conventional $\Sigma\Delta$ loop is always active ($+1$ or $-1$), whereas the asynchronous model is only active at the time of a spike event, making the model more energy-efficient.

The input signals to the neuron may be encoded as spikes trains or as continuous analogue values from a sensor. The transmitted analogue signal is reconstructed from a spike train by simply low-pass filtering it. An advantage of the $\Sigma\Delta$ neuron model is that it comes with a low-pass filter in its input stage. Therefore, a $\Sigma\Delta$ neuron is a "codec" - It can both encode an analogue signal into a spike train and decode an incoming spike train back to the transmitted analogue signal. As the low-pass filter at the input stage is agnostic of the type of input to it, a $\Sigma\Delta$ can encode and decode both types of signals - spike trains or continuous analogue ones. A description of the biological motivation and noise-filtering properties of the model is provided in supplementary section A. The circuit implementing this model has been fully characterized in Nair and Indiveri (2019).

## 4    TRAINING A SPIKING RNN WITHOUT SIMULATING SPIKES

The neuron model introduced in the previous section allows us to train a recurrent SNN by treating the spiking neurons as LPFs and modifying the recurrent ANN equations suitably. We will show that the trained weight parameters of the recurrent ANN model can be mapped to a recurrent SNN, without additional training. We measure the effectiveness of this mapping procedure by comparing the temporal dynamics of the neurons in the recurrent ANN to the low-pass filtered spike trains generated by the spiking neurons in the corresponding recurrent SNN. In this demonstration, we use high precision synaptic weights. This is typically not available in most spiking neuromorphic platforms. However, the same mapping procedure can be used for mapping ANNs trained with binary or noisy weights. Before introducing the mapping procedure, we describe three operations that are needed for it.

**Input re-scaling:** When implementing an SNN in mixed-signal neuromorphic systems, the state variables of the neuron are represented by voltages or currents that are of the order of mV or nA. Using such small values when training an ANN in software may lead to computational instability. To avoid this we train the network with normalized input signals and re-scale the parameters and activation functions after training. For example, if a single layer calculation is represented as $y = \sigma_{nl}(W \cdot x)$, where $\sigma_{nl}$ is a non-linear activation function, $x$, $W$ and $y$ are the inputs, weights, and outputs from the layer, respectively, then, to re-scale the inputs by a factor $\gamma$, the activation function used in the ANN will be modified, during inference, as $y = \overline{\sigma_{nl}}(W \cdot \gamma \cdot x)$, where, $\overline{\sigma_{nl}}(.) = \sigma_{nl}\left(\frac{.}{\gamma}\right)$

**Low-pass filtering:** The input signal can be reconstructed from the spikes trains generated by the $\Sigma\Delta$ neuron if $s(t)$ tracks $i(t)$ (Equation 4). This is because $s(t)$ is obtained by filtering the neuron spike train. This is why a $\Sigma\Delta$ neuron can be modelled as an LPF with time constant $\tau_w$. The LPF approximation ignores the high-frequency components injected by the spiking mechanism, as they

are suppressed by feedback loop (the $N(s)$ term in the transfer function of $\Sigma\Delta$ neuron, Equation 5). We model this by using a discrete-time Euler approximation to incorporate a LPF-term at the output of the RNN stage. This results in the Low-Pass Recurrent Neural Network (lpRNN) cell by a simple tweak to the classical equation:

$$y_t = \alpha \odot y_{t-1} + (1 - \alpha) \odot \sigma(W_{rec} \cdot y_{t-1} + W_{in} \cdot x_t + b) \qquad (6)$$

where, $\sigma$, $\odot$ and $\cdot$ denote non-linearity, element-wise Hadamard product and matrix multiplication functions, respectively. The variables $\alpha$, $y_n$, $x_n$, $W_{rec}$, $W_{in}$, and $b$ represent the retention ratio vector, input vector, output vector, recurrent connectivity weight matrix, input connectivity weight matrix, and biases, respectively. The subscripts on variable $y$ and $x$ indicate the time step. $\alpha$ models the time constant of the recurrent ANN, and it is matched to the SNN time constant by setting it to $\alpha = e^{\frac{-Ts}{\tau_s}}$ where, $Ts$ is the time-step of the input data-stream fed to the recurrent ANN, and $\tau_s$ is the feedback time constant of the $\Sigma\Delta$ neurons used in the desiredSNN.

For example, if the recurrent ANN is being trained to detect speech from an audio-signal, then $Ts$ should be set equal to the time difference between the consecutive samples. The value of $\tau_s$, for the ANN set to $min(\tau_w, \tau_{mem})$ in the $\Sigma\Delta$ equations. $\tau_s$ must therefore be chosen such that the signals being transmitted lie well within the pass-band of the feedback filter. This ensures that all the in-band components are transmitted well, even when different neurons in the systems have different values of $\tau_s$, for example, due to mismatch. This is an important observation for mixed-signal systems, where mismatch effects may result in different neurons to have differing time-constants. We will demonstrate that the effect of device mismatch is well-tolerated for most practical cases and leads to gradual degradation in performance as it increases. It must be noted that while computing $Ts$ or the bandwidth of an audio or sensor measurement is easy, it is not trivial for data-sets such as text.

**Saturating non-linearity:** It has already been shown in the literature that the Rectified Linear Unit (ReLU) non-linearity is a good non-linear model of the aI&F neuron (Yoon, 2016; Bohte, 2012). However, the $\Sigma\Delta$ neuron also filters incoming spike trains using a low-pass filter which limits its maximum output current to $I_{in}$ (see Equation 4). To model this effect we can either set $I_{in}$ in the SNN to the largest activation output found in the ANN simulation or clamp the maximum output of the activation function in the ANN simulation to $I_{in}$. In our experiments, we do the latter.

### 4.1 THE MAPPING PROCEDURE

First, the recurrent ANN cell is modified by replacing the RNN units with the lpRNN. The modified network is then trained using Backprop with conventional Autograd tools provided by libraries such as PyTorch or Tensorflow. This gives us the synaptic weights for the recurrent SNN. Then, the largest value attained by the state variables in the trained network is mapped to $I_{in}$. This ensures that the spiking neurons do not saturate. Finally, the inputs to the SNN are re-scaled to suitable currents or voltage values as described earlier.

**Limitation:** A recurrent SNN is a continuous-time system that spikes hundreds to thousands of times per second to achieve the necessary transmission accuracy. Therefore, the time step of the transient simulation needs to be made very fine. The mapping algorithm assumes that the mapped SNN operates on the same sequence that the original ANN is trained on. This is a problem. Training a recurrent ANN on a long sequence using back-propagation is computationally expensive and often intractable because of vanishing and exploding gradients. For example, with speech signals sampled at the standard rate of 44.1 kHz, even a short utterance is thousands of samples long. If we are unable to train an ANN for the desired task, the mapping mechanism is useless. Our approach to addressing this issue is to train the ANN with sub-sampled signals. After we compute the desired weights, we rescale the time-constants of the network before mapping it to the SNN. If the simulation time-step for the SNN is $T_{s_{SNN}}$ and that of ANN is $T_{s_{ANN}}$, then the time constants of the two simulations are given by $\alpha_{ANN} = e^{-\frac{T_{s_{ANN}}}{\tau}}$ and $\alpha_{SNN} = e^{-\frac{T_{s_{SNN}}}{\tau}}$. We only rescale the time constants without changing the weights of the mapped network, introducing inaccuracies in the mapped network. The mismatch arises because a single time-step of the ANN corresponds to several simulation time-steps in the mapped SNN ($= \frac{T_{s_{ANN}}}{T_{s_{SNN}}}$). The mapping is exact for a first-order LPF because of the Linear Time-Invariant (LTI) property. However, even though an RNN is non-linear, by making the low-pass filtering effect more dominant (for example, with $\alpha = 0.99$), we observe that the mapped dynamics match well.

## 5 Experimental results

All the spiking simulations in the following sections are run on a custom transient mixed-signal modeling library, called spiking simulator for systems of 1st-order LPFs (Spiker). The motivation for design and operation of Spiker is described in supplementary section E.

### 5.1 Encoding performance of the Sigma–Delta neuron model

The $\Sigma\Delta$ neuron model used in the mapped SNN is not an ideal transmitter of information as the spiking mechanism introduces error akin to quantization noise. This is analogous to use of low bit-precision in conventional ANNs. It is important to measure how much precision is available using a metric that is meaningful for the proposed spiking architecture. We measure this using a Signal-to-Distortion ratio (SDR) metric when encoding a sinusoidal input and reconstructing it with an LPF. The SDR is the ratio between the energy contained in the transmitted signal and total energy in all other frequency components generated by the distortions introduced in the signal chain. The results of these experiments are shown in Figure 2. We note in Figure 2a that highest SDR of the $\Sigma\Delta$ neuron is 55dB, with a 20 dB/decade roll-off as a function of frequency with a pole corresponding to $\tau_{mem}$. Figure 2b highlights the input amplitude-dependence of the SDR. We note in Figure 2b that the SDR improves as a logarithmic function of the input amplitude and then drops suddenly. The logarithmic improvement in SDR is because the error component corresponding to the spiking threshold, $\Delta$, becomes a smaller fraction of the input amplitude. The sudden drop occurs when the input amplitude approaches and exceeds $I_{in}$ in Equation 4. This is because the maximum attainable value of the feedback term, $s(t)$ in Equation 4 is $I_{in}$. Under these conditions, $I_{mem}$ is always greater than $s(t)$, causing the neuron to fire at a very high rate. For the rest of the SNN simulations in this paper, the $\Sigma\Delta$ neuron settings listed in Figure 2 caption are used.

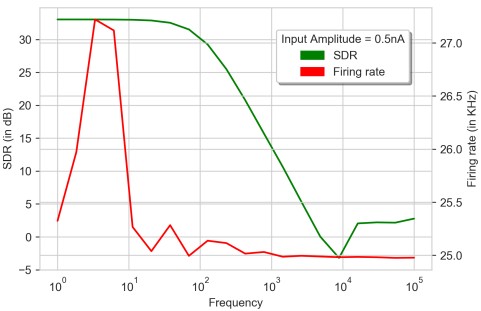

(a) SDR vs input sinusoid frequency.

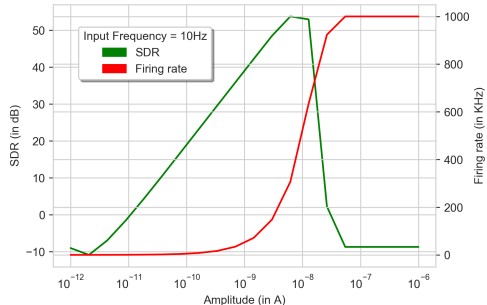

(b) SDR vs input sinusoid amplitude.

Figure 2: SDR of the $\Sigma\Delta$ neuron as a function of sinusoidal input parameters. The $\Sigma\Delta$ neuron is fed a single-tone sinusoid riding on a DC bias to ensures that the input is non-negative. The transient simulations are run with a simulation time step of $1\mu s$. This is the reason for the saturation in the firing rate in (b). The SDR ratio is reported after subtracting the DC component. The neuron parameters for the simulation are $\tau_{mem} = 0.007s$, $\tau_w = 0.0014s$, $\alpha_L = 5000$, $\alpha_s = 1$, $\Delta = 0.1nA$, $I_{in} = 40nA$, $I_L = 0nA$.

### 5.2 Demonstration of the mapping mechanism

We demonstrate the mapping mechanism using multi-layer RNNs. The ANN dynamics are compared against a signal obtained by low-pass filtering the spike trains generated by the $\Sigma\Delta$ neuron. Our assertion is that the mapping mechanism will map any recurrent ANN to an equivalent recurrent SNN. Therefore, instead of demonstrating the mapping for a particular task, we set synaptic weights to random samples from a Gaussian distribution. We then compare the dynamics of all the neuron units in the mapped and original networks. To ensure that our nodes do not saturate, we constrain the largest eigenvalue of the recurrent weight matrices to 1.4. The motivation for this trick was from obtained from Echo-State Networks (ESNs)(Jaeger, 2002; Jaeger et al., 2007). The quality or goodness of fit is measured using an Normalized mean square error (NMSE) metric, which measures

the mean square error normalized by the signal power. It is used to compare two time series signals $x_{ref}$ and $x$ using the following formulation:

$$NMSE = 1 - \frac{||x_{ref} - x||^2}{||x_{ref} - mean(x_{ref})||^2} \qquad (7)$$

where, $||.||$ indicates the L2 norm. The $NMSE$ metric lies between 1 and $-\infty$, with 1 indicating a perfect match and $-\infty$ indicating a very bad fit. If $NMSE = 0$, then x is at least as good a fit as a straight line at $x_{ref}$.. In our results, we report the mean and standard deviation in the NMSE scores for all the units in a layer.

The effectiveness of the mapping mechanism is demonstrated using a four-layer RNN. The input and output stages are implemented as fully-connected feed-forward layers, and the recurrent layers are also interleaved with fully-connected layers. The input feature dimension was set to two and the output to three. The input data was a weighted sum of sinusoidal signals that were band-limited to 50 Hz and sampled at 1 MHz for 0.2 seconds. The high sampling rate is necessary to accurately capture the dynamics of the SNN, whose neuron models are highly non-linear, in a transient simulation. The length of the simulation is a key consideration as we want our mapped SNN implementation to remain matched for arbitrarily long sequences. Computational considerations limited the duration of our simulations, but this should be tested before large scale deployment in real-world use.

Fully-digital neuromorphic platforms, such as Intel Loihi, IBM TrueNorth, or SpinNaker, do not suffer for mismatch issues. However, mixed-signal neuromorphic chips, such as Qiao et al. (2015); Neckar et al. (2019); Schemmel et al. (2012), are potentially more energy-efficient than their digital counterparts but suffer from device mismatch. A $\Sigma\Delta$ feedback loop naturally compensates for such effects (Pavan et al., 2017) but there are many components in the model that lies outside the feedback loop. To study this, we add the effect of mismatch in our simulations by sampling the parameters, $p$ of the mapped SNN:

$$p = p \cdot (1 + c_{v_p} \cdot \mathcal{N}(0,1)) \qquad (8)$$

where, $c_{v_p}$ is the coefficient of variation ($= \frac{standard\ deviation}{mean}$) in the parameter, $p$. To understand the statistics in the quality of the mapping mechanism, we generate multiple samples of network parameters and measure the quality of fit. These results are tabulated in Tables 1 and 2, where we list the measured mean of and standard deviation in $NMSE$ values for a 4-layer RNN with 51 and 500 units per layer, respectively. With no mismatch effects, the reconstruction is very good to all layers, in both cases. Furthermore, we observe nearly perfect reproduction of the network dynamics for up to 2 layers, and a gradual degradation as the size, depth and mismatch of the network increases. We note that the mapping mechanism is robust for $c_{v_p} < 0.2$. Reduced mismatch sensitivity is useful for design of neuromorphic chips because it simplifies the design, and that, in turn, reduces the energy and area consumed by these chips. Visualization of the transient dynamics of all the nodes in the original and mapped RNNs is provided in supplementary section F. Finally, the performance of the mapping algorithm comparing the dynamics of the ANN with sub-sampled data to that of the SNN is shown in Table 3. The length of the SNN simulation is $0.2s$, translating to input sequence lengths, L. We note that the mapping technique works well for fairly high sub-sampling ratios and shows significant degradation only for $L = 20$. Note that a near perfect match ($NMSE > 0.5$) is achieved up for depth of two. This restricts the models used for benchmarking in Section 5.3.

## 5.3   LEARNING PERFORMANCE OF THE LPRNN CELL

The key idea behind enabling the mapping between an recurrent ANN to its spiking equivalent was the addition of a low pass filter to the the state variables. It must be highlighted that the idea of using a low-pass filter model for neurons is fairly old and has been studied in various contexts (Beer, 1995; Mozer, 1992; Jaeger et al., 2007). The novelty in this work is in identifying its use in the mapping mechanism and in the study of its learning properties. The mapping procedure would be of no use if the resulting ANN was unable to perform as well as their unfiltered counterparts in learning tasks, and it is the focus of this section.

In our experiments, we set $\alpha$ in Equation 6 by sampling from a distribution with a common mean value shared by all the neuron units in the network. This simplifies the design of the neuromorphic system by eliminating the need to create precise tunable time constants in the neuron implementations.

| Layer | $\mu_{NMSE}$ | | | | $\sigma_{NMSE}$ | | | |
|---|---|---|---|---|---|---|---|---|
| | $c_{v_p} = 0$ | $c_{v_p} = 0.2$ | $c_{v_p} = 1$ | $c_{v_p} = 2$ | $c_{v_p} = 0$ | $c_{v_p} = 0.2$ | $c_{v_p} = 1$ | $c_{v_p} = 2$ |
| Rec. layer 1 | 1.0 | 0.9 | -5.1 | -4.0 | 0.1 | 0.3 | 55.1 | 15.5 |
| Rec. layer 2 | 1.0 | 0.5 | -8.8 | -17.5 | 0.1 | 1.1 | 54.4 | 41.4 |
| Rec. layer 3 | 0.9 | -1.5 | -36.8 | -100.2 | 0.1 | 6.4 | 117.1 | 300.0 |
| Rec. layer 4 | 0.9 | -3.7 | -107.5 | -278.9 | 0.2 | 8.2 | 202.4 | 535.9 |

Table 1: Mapping a four layer network with 51 units per layer for different mismatch values.

| Layer | $\mu_{NMSE}$ | | | | $\sigma_{NMSE}$ | | | |
|---|---|---|---|---|---|---|---|---|
| | $c_{v_p} = 0$ | $c_{v_p} = 0.2$ | $c_{v_p} = 1$ | $c_{v_p} = 2$ | $c_{v_p} = 0$ | $c_{v_p} = 0.2$ | $c_{v_p} = 1$ | $c_{v_p} = 2$ |
| Rec. layer 1 | 1.0 | 1.0 | 0.4 | 0.6 | 0.1 | 0.1 | 1.8 | 1.3 |
| Rec. layer 2 | 0.9 | -1.0 | -48.9 | -132.0 | 0.1 | 1.6 | 52.1 | 143.7 |
| Rec. layer 3 | 0.8 | -6.8 | -185.1 | -683.6 | 0.2 | 2.9 | 73.2 | 295.9 |
| Rec. layer 4 | 0.2 | -6.3 | -133.7 | -416.3 | 0.9 | 3.5 | 64.0 | 203.6 |

Table 2: Mapping a four layer network with 500 units per layer for different mismatch values.

| Layer | $\mu_{NMSE}$ | | | | $\sigma_{NMSE}$ | | | |
|---|---|---|---|---|---|---|---|---|
| | $T_{s_{ANN}} = 10\,\mu s$ $L = 20000$ | $100\mu s$ $2000$ | $1ms$ $200$ | $10ms$ $20$ | $10\,\mu s$ $20000$ | $100\mu s$ $2000$ | $1ms$ $200$ | $10ms$ $20$ |
| Rec. layer 1 | 1.0 | 1.0 | 0.8 | -0.4 | 0.0 | 0.0 | 0.1 | 0.4 |
| Rec. layer 2 | 0.9 | 1.0 | 0.7 | -1.7 | 0.8 | 0.0 | 0.2 | 5.2 |
| Rec. layer 3 | 0.9 | 0.9 | 0.4 | -1.6 | 0.5 | 0.1 | 1.0 | 2.2 |
| Rec. layer 4 | 0.9 | 0.9 | 0.1 | -2.0 | 0.3 | 0.3 | 1.2 | 2.2 |

Table 3: Performance when mapping resampled data for a four layer network with 128 units per layer.

| Task | Results in this work | | | Reference literature |
|---|---|---|---|---|
| | SimpleRNN | lpRNN | LSTM | |
| Temporal addition | 40 steps | **642 steps** | $\infty$ | 5000 steps (Li et al., 2018) |
| Temporal copy | 30 steps | **120 steps** | 200 steps | 500 steps (Arjovsky et al., 2016) |
| Spoken phrase (acc.) | 27% | **93%** | 92% | 90% (Sainath and Parada, 2015) |

Table 4: The performance of the lpRNN cell on three benchmarks. The reference literature column reports the best results (to our knowledge) with neural networks with less than 100K parameters. lpRNN and SimpleRNN networks for the add and copy tasks used a single layer with 128 hidden units. Arjovsky et al. (2016) use unitary recurrent weight matrices and Li et al. (2018) use two layers of 128-unit IndRNN layers. The spoken phrase task reference is a CNN model released as a Tensorflow example (Google, 2019).

First, we study the short-term memory capabilities of the lpRNNs using the synthetic addition and copying tasks (Hochreiter and Schmidhuber, 1997; Arjovsky et al., 2016; Le et al., 2015). Next, we compare the performance of the lpRNN cell vs a SimpleRNN cell in a speech recognition task. We summarize the key observations in this section and details are in the supplementary material.

**Temporal addition task:** The addition task (Le et al., 2015) involves processing two parallel input data streams of equal length. The first stream comprises random numbers $\in (0, 1)$ and the second is full of zeros except at two time steps. The network is trained to generate the sum of the two numbers in the first data stream corresponding to the time-steps when the second stream had non-zero entries. The baseline to beat is a mean square error (mse) of 0.1767, corresponding to a network that always generates 1. We adopt a curriculum learning (Bengio et al., 2009) procedure for this task, by first training the RNN cell on a short sequence and progressively increasing the number of time steps. Each curriculum used 10,000 training and 1000 test samples. The length of the task was incremented when the mse went below 0.001. The SimpleRNN cell failed to converge beyond sequences of length 40, while the lpRNN cell converged to mse $< 0.001$ for sequences up to 642 steps. Interestingly, the

Long Short-Term Memory (LSTM) cell learnt a general solution when trained by this procedure and could solve arbitrarily long sequences, even with just two hidden units in the cell!

**Temporal copying task:** We train the RNN cells on a varying length copying task as defined in (Graves et al., 2014) instead of the original definition (Hochreiter and Schmidhuber, 1997; Arjovsky et al., 2016). This problem is harder to solve than the temporal addition task. The network receives a sequence of up to S symbols (in the original definition, S is fixed) drawn from an alphabet of size K. At the end of S symbols, a sequence of T blank symbols ending with a trigger symbol is passed. The trigger symbol indicates that the network should reproduce the first S symbols in the same order. We adopt a curriculum learning procedure here too and first train the network on a short sequence (T=3) and gradually increase it (T$\leq$200). The sequence length is incremented when the categorical accuracy is better than 99%. The SimpleRNN cell failed at this task even for T=30. The lpRNN cell was able to achieve 99% accuracy for up to 120 time steps. After that, it generates T blank entries accurately but the accuracy of the last S symbols drops (For T=200, it was 96%).

**Spoken phrase classification:** The target for the lpRNN cell are neuromorphic platforms that are typically resource-constrained, due to power, memory, and area restrictions. Therefore, we test the performance of the lpRNN cell on a problem that is compatible with such systems (see supplementary section D). We chose a limited vocabulary spoken commands detection task using the Google commands dataset (Warden, 2018), which comprises 36 classes of short-length spoken commands such as "left", "up", or "go". The dataset was created for hardware and algorithm developers to evaluate their low footprint neural network models in limited dictionary speech recognition tasks. The neural network receives a Mel-spectrogram as inputs, and comprises, from input to output, two fully-connected dense layers with 128 and 32 units with batch normalization, two RNN layers with 128 units, two fully-connected dense layers, topped by a softmax readout. In total, the network has about 80K trainable parameters. We use Adam optimizer (Kingma and Ba, 2014), with a learning rate of 0.001 for training. We note that all lpRNN variants massively outperform the SimpleRNN and slightly outperform the LSTM variants, all with the same number of parameters. We further note that the performance of the network peaks for certain values of $\alpha$ and the random sampling case. The high performance of the lpRNN cell is a crucial result because not only did the addition of the low-pass filter enable mapping of the recurrent ANN model to neuromorphic platforms, it also resulted in a much-improved performance.

| $\alpha$ | [0.1,1] | 0 | 0.1 | 0.5 | 0.8 | 0.9 | 0.99 | 1 |
|---|---|---|---|---|---|---|---|---|
| **Accuracy** | **93%** | 26% | 25% | 69% | 92% | 93% | 90% | 4% |

Table 5: Performance of the lpRNN cell on the Google commands classification task with different filtering coefficients, $\alpha$. $\alpha = 0$ is equivalent to a SimpleRNN and $\alpha = 1$ is an MLP.

**Energy-efficiency estimation:** We compare the energy cost of computing a two-layer RNN with 128 hidden units (used in the audio task) for an in-memory lpRNN system with $\Sigma\Delta$ neurons against Cortex-M4, a popular low-power microprocessor used in milliWatt-range applications. We conservatively estimate that our implementation yields $500\times$ better energy-efficiency in this instance. Larger networks will see a dramatic increase in this difference. (see supplementary section J).

## 6 CONCLUSION AND OUTLOOK

We presented a novel aI&F spiking neuron model and discussed its spike-coding and noise-tolerance benefits. Then, we discussed how the LPF abstraction of the aI&F neuron model enables training of recurrent SNNs without having to simulate the complex dynamics of a large network of spiking neurons. This strategy enables training a recurrent SNN using standard optimization algorithms such as backpropagation. The mapping technique that allowed us to achieve this result is studied in detail, including its limitations.. We observe a dramatic improvement in the performance of the RNN cell with the addition of the low-pass filtering term. We think that this improvement is because the LPF acts as a temporal regularizer (see supplementary section C). Current spiking RNNs are benchmarked on much simpler tasks than presented in this paper, often due to the computational cost and lack of algorithms for training them. The importance of this work lies in proposing algorithmic solutions to address this key problem, to enable a new generation of ultra-low-power neuromorphic chips.

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

# A    MOTIVATION FOR THE USE OF THE SIGMA-DELTA NEURON MODEL

## A.1    CODING MECHANISM

A neuron in deep learning has one primary function - the non-linear transformation of its input. However, a biological neuron and its neuromorphic counterpart have the added job of encoding information in spike trains. The most popular model for this encoding mechanism is rate coding, where the neuron fires at a rate proportional to the incoming signal. This mechanism is not efficient for transmitting high-resolution data. For example, to transmit a signal at 8-bit resolution, it will require O(256) spikes for each sample. An improvement to this coding mechanism is theaI&F model (Brette and Gerstner, 2005), which can be interpreted as an asynchronous delta-sigma ($\Delta\Sigma$) loop (Yoon, 2016; Bohte, 2012; Nair and Indiveri, 2019). The $\Delta\Sigma$ mechanism is a time-coding model that is more efficient in its use of spike trains (Nair and Indiveri, 2019) than rate-coding models. Other time-coding schemes have also been proposed in literature (Rueckauer et al., 2017; Gerstner and Kistler, 2002). However, the advantage of the $\Delta\Sigma$ interpretation is that it leads to highly power-efficient circuit implementations (Nair and Indiveri, 2019) that is tolerant to mismatch and noise effects. Furthermore, the model allows us to treat the neuron state as an analogue variable and ignoring the specific timing details of the encoding spike trains (Nair and Indiveri, 2019). The independence from the monitoring precise spike-times is beneficial because state-dependency, noise and device mismatch cause different neurons to generate spikes at different times for the same input. Modelling it is computationally expensive. The $\Delta\Sigma$ feedback loop ensures that they all represent the same signal with the same accuracy in spite of their differences (Nair and Indiveri, 2019). This abstraction enables the network designer to only look at the internal state of the neuron when optimizing the network weights for an SNN. It is a crucial enabler for this paper as simulating and training mismatch-prone SNNs is computationally much more expensive than ANNs.

## A.2    BIOLOGICAL NEURAL NETWORKS ARE LOW PASS FILTERING

Activation functions used in neural networks and apply a non-linearity to a weighted sum of input signals. However, ANNs assume that when the input changes, the internal state of the neuron or dendrites can also change immediately to reflect the new input. This behaviour ignores the fact that biological neuronal channels are LPFs (Gerstner and Kistler, 2002). Modelling the inertial or low-pass filtering property is essential to implement and study recurrent neural networks in any neuromorphic system as the transitional dynamics deviate completely in its absence. The $\Delta\Sigma$ neuron models the filtering behaviour with a first-order low pass filter. We argue that not only is this modelling essential, it is also a useful constraint to impose on RNNs.

## A.3    CHANNEL NOISE AND RECONSTRUCTION ACCURACY IN A SPIKING NEURON

The $\Sigma\Delta$ scheme encodes the information in the relative timing of the spikes. This implies that any noise in the spike timing, i.e. jitter, introduced during transmission of the spikes will result in distortion of the reconstructed signal, for example, in situations when the transmitting and receiving neurons are on separate chips. The distortions are modelled by a random variable $\Delta_t$, which is sampled from a normal random distribution, $\Delta_t \sim \mathcal{N}(0, j_\sigma^2)$ with probability distribution function $Pr(\Delta_t)$. Note that a fixed delay does not contribute to distortion and is ignored. To compute the effect of spike timing noise, we calculate the Laplace domain representation of the transmitted signal as follows. If $I_D(t)$ is the Dirac delta function, the transmitted spike train $y(t)$, jitter-affected spike train, $y'(t)$, the desired filtered spike train, $s(t)$, and the filtered version of the jitter-affected spike-train,

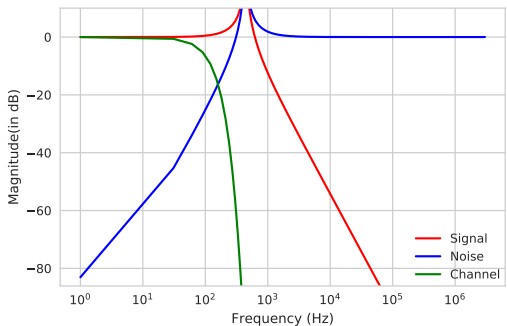

Figure A.1: The different transfer function of the $\Sigma\Delta$ neuron model. Red: Signal transfer function, Blue: noise transfer function, Green: Signal transfer function with spike timing noise.

$r(t)$ can be expressed as,

$$y(t) = \sum_{k=0}^{N} I_D(t - t_k) \tag{9}$$

$$y'(t) = \sum_{k=0}^{N} I_D(t - t_k - \Delta_t) \tag{10}$$

$$s(t) = y(t) * h(t) = \sum_{k=0}^{N} h(t - t_k) \tag{11}$$

$$\implies S(s) = \sum_{k=0}^{N} H(s)e^{-st_k} \tag{12}$$

$$r(t) = y'(t) * h(t) = \sum_{k=0}^{N} h(t - t_k - \Delta_t) \tag{13}$$

The mean value of $r(t)$ can then be computed as

$$\bar{r}(t) = E[r(t)] = \sum_{k=0}^{N} \int_{\Delta_t=-\infty}^{\infty} Pr(\Delta_t) \cdot h(t - t_k - \Delta_t)$$

$$= \sum_{k=0}^{N} Pr \star h(t - t_k) \tag{14}$$

$$\implies \overline{R}(s) = \mathscr{L}\{\bar{r}(t)\} = \sum_{k=0}^{N} P(s)H(s)e^{-st_k} = P(s)S(s) \tag{15}$$

where, $P(s) = \mathscr{L}\{\Pr(t)\} = e^{\frac{s^2 j_\sigma^2}{2}}$. $\overline{R}(s)$ is plotted in Figure A.1 as the recovery transfer function (RTF) with $j_\sigma = 1ms$. We see that for sufficiently band-limited signal, the channel noise does not affect the signal transmission accuracy.

## B    COMPARISON TO OTHER RNN MODELS

The low pass filtering behaviour of neurons is well-known in neuro-scientific literature (and in other fields) and has been studied in the past with RNNs as well(Beer, 1995; Mozer, 1992; Jaeger et al., 2007). For example, ESNs as proposed by Herbert Jaeger (Jaeger et al., 2007) has an identical formulation to the lpRNN where the recurrent layer uses leaky integration units. In ESNs, the spectral radius of the initialization values of the recurrent kernel is constrained to confers an "echo-state

property" to the network. The recurrent or input connectivity weights are not trained during the learning process. Instead, only the read-out linear classifier is trained. In the lpRNN model, the spectral radius of the recurrent kernel is not constrained and all the weight matrices, including the retention ratios if required, are trained. lpRNN also shares similarities with recurrent residual networks proposed by Yiren Wang (Wang and Tian, 2016), which are described by the following equations

$$y_t = f(g(y_{t-1})) + \sigma(y_{t-1}, x_t, W) \tag{16}$$

where $W$ denotes input and recurrent kernels, and other symbols have the same meaning as equation 6. In equation 16, $g$ and $f$ are identity and a hyperbolic tangent functions, respectively. A comparison can also be made with the LT-RNN model proposed by Mikael Henaff (Henaff et al., 2016), whose update equations are:

$$h_t = \sigma(W_{in} \cdot x + b) + V \cdot h_{t-1} \tag{17}$$
$$y_t = W \cdot h_t \tag{18}$$

where $W$ and $V$ are 2-D transition matrices that are learned during the training process. In this case, it is possible that the LT-RNN cell reduces to an lpRNN, but is unlikely to occur in practice. Similar analogies can also be made to the IndRNN model (Li et al., 2018) and recurrent identity networks (Hu et al., 2018). Generally speaking, the main difference between the lpRNN cells and popular RNN models in use today is the(re)indroduction of the filtering term into the RNN model with impositions on boundary and train-ability conditions. We see that in addition to enabling their use in neuromorphic platforms, this results in more stable convergence properties due to a temporal regularization effect, as described in the Section C.

## C  MEMORY IN AN LPRNN

We can analyze the evolution of an lpRNN cell by using an approach similar to the power iteration method, described by Razvan Pascanu (Pascanu et al., 2013). To do this analysis, we approximate the lpRNN update equation as:

$$y_t = \alpha \odot y_{t-1} + (1 - \alpha) \odot (W_{rec} \cdot y_{t-1} + W_{in} \cdot x_t + b) \tag{19}$$

where, for simplicity, we also make the added assumption that all units of the lpRNN layer have the same retention factor, $\alpha$. The gradient terms during back-propagation through time can now be expressed as a product of several terms that have the form:

$$\frac{\delta y_t}{\delta y_k} = [(1 - \alpha)W_{rec}^T + \alpha]^l \tag{20}$$

where, $t$ and $k$, are time step indices with $t > k$ and $l = t - k$. If an eigenvalue of the $W_{rec}^T$ matrix is $\lambda$, then the corresponding eigenvalue of the matrix $[(1 - \alpha)W_{rec}^T + \alpha]$ can be written as

$$(1 - \alpha)\lambda + \alpha \tag{21}$$

Looking at the eigenvalue of the gradient terms as computed in equation 21, we note that $\alpha$ acts like a **temporal regularizer** on the eigenvalues of the recurrent network. It can also be seen that by scaling $\alpha$ to lie between 0 and 1, the operation of the network shifts between that of purely non-inertial recurrent to a completely inertial network stuck in its initial state, respectively. This insight helps us understand why lpRNNs perform well in long memory tasks.

Hochreiter (Hochreiter and Schmidhuber, 1997) defined the constant error carousel (CEC) as a central feature of the LSTM networks that allowed it to remember past events. In a crude sense, this corresponds to setting the retention ratio, $\alpha = 1$. Forget gates were subsequently added by Felix A. Gers (Gers et al., 2000) to the original LSTM structure, that allowed the network to also erase unnecessary events that were potentially trapped in the CEC. This means that the average effective weight of the self-connection in the CEC was made $< 1$. A randomly initialized set of $\alpha$ values with a reasonably large number of cells appears to have similar functionality. By setting $\alpha < 1$, the network is guaranteed to lose memory over time, but if some of the $\alpha$s are close to 1, it may retain the information for a longer time frame. Moreover, the regularization effect of $\alpha$s also prevents the eigenvalues of the recurrent network from becoming too small, ensuring that memory is never lost immediately. We expect that the lpRNN model has a reduced representational power than gated RNN cells, not simply because it has 4x fewer parameters, but because the lpRNN state is guaranteed to fade with time whereas a gated cell can potentially store a state indefinitely.

## D  THE NEUROMORPHIC SIGNAL CHAIN

The $\Delta\Sigma$ mapping mechanism requires defining suitable time constants for the lpRNN cell being trained by backprop. This can be derived for continuous-time signals from sensors or real-world signals such as an audio input by taking into account the bandwidth of the incoming signals as described earlier. We illustrate a reference neuromorphic signal chain for processing audio input in Figure D.2. The data received from the audio sensor is first filtered by an audio filtering stage such as the cochlea chips (Chan et al., 2007; Sarpeshkar, 1998; Hamilton et al., 2008). These systems typically implement mel-spaced filter banks. A neural network processes the filter outputs and drives an actuator system. A minimal configuration of weights and connectivity required to implement the lpRNN cell in a neuromorphic platform is illustrated in Figure D.2. It is a memory array with spiking neurons attached to the periphery of the system. Each memory cell acts as a transconductance stage - it receives voltage spikes and generates a scaled output current. These currents are summed by the Kirchoff's current law and integrated by the neurons. Readers familiar with memory design and computer architecture may identify this as an in-memory computational unit. An in-memory neural network accelerator is energy-efficient, primarily because it eliminates movement of synaptic weights (Ghose et al., 2018; Qiao et al., 2015) from the memory to a far-away processing module. Instead, the activations of the neurons are transmitted to the other nodes in the network. The computation is no longer memory-bound unlike RNN computation on von Neumann style architecture. Figure D.2 implements a single spiking RNN stage (equivalent to an lpRNN), with green and red boxes highlighting the input and recurrent kernels, respectively. The architecture can be modified to implement a fully connected layer by eliminating recurrent connections. Note that an equivalent configuration can be also set up in fully-digital neuromorphic systems such as (Frenkel et al., 2019; Davies et al., 2018; Akopyan et al., 2015) that do not suffer from noise and mismatch issues, but may consume more area and power.

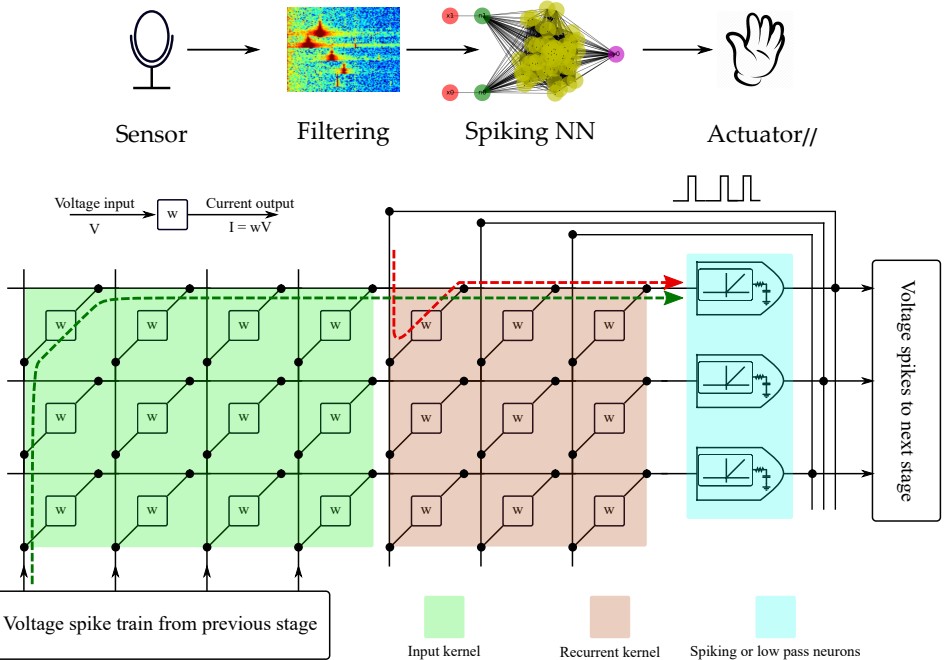

Figure D.2: Top: A neuromorphic signal chain. Bottom: Architecture of an SNN accelerator implementing an lpRNN.

## E  SPIKER : A SYSTEM-LEVEL SPIKING SIMULATOR

Spiker is a transient-simulator for simulating large networks of spiking neurons and synapses using the Basic Linear Algebra Subprograms (BLAS) libraries. It is written in Python and uses the

Numpy (Oliphant, 2006) library. There is also a PyTorch (Paszke et al., 2017) version that supports Graphical Processing Unit (GPU)-acceleration. Unlike spiking simulation tools like Brian2 (Stimberg et al., 2019) or NEST (Gewaltig and Diesmann, 2007), which are general-purpose solvers of Ordnary Differential Equations (ODEs), Spiker is a highly-specific simulator for systems where the only differential equation implemented is that of a first-order LPFs. The simulator is not designed to solve any differential equation. Instead, it allows us to run massive simulations of large networks comprising neuron and synapse models whose building blocks include first-order LPFs.

### E.1    How does Spiker work?

**Key idea #1**    Constraining the support to first-order LPFs has the advantage that it allows us to create closed-form solutions to the differential equations at all time-step resolutions without loss of accuracy. The differential equation corresponding to a first-order LPF is the following:

$$\tau \frac{dx}{dt} = -x \tag{22}$$

The closed-form solution to this, ignoring the initial conditions takes the form

$$x = x_0 \cdot e^{\frac{-t}{\tau}} \tag{23}$$

A useful property of the exponential function is that $e^{-t1+t2} = e^{-t1} \cdot e^{-t2}$. Therefore, we have a simple closed-form method to compute the state of a LPF at any arbitrary time, given an initial condition. This implies that if all the building blocks of a system are made of modules which have an exponential solution, then, given their initial state, it is possible to compute the state of the entire system at some arbitrary time in the future, precisely. This is a well-known property of all LTI systems.

**Key idea #2**    However, a spiking system is not LTI. Therefore, it is not possible to predict the state of a spiking neural network at an arbitrary time in the future. Instead, the Spiker simulator takes tiny temporal steps and computes the state of the network variables at each step using the closed-form solution. At the end of each time-step, the neuron models check if it should spike and then resets the corresponding variable to zero. The spiking input to the synapses is a train of 1-bit values corresponding to the presence or absence of an incident spike from an upstream neuron. This implies that the precision of the spike-event is limited to the resolution of the simulation time-step. However, the key insight here is that, if the signals being transmitted have a bandwidth much smaller than the simulation time-step, the higher-order effects can be safely assumed to be gone.

Moreover, in spiking neurons, small amounts of jitter in the spike-timing is well-tolerated. This is because of the noise-cancelling property of the feedback loop. Therefore, the Spiker simulator allows the user to set the simulation time-step to a large value that offers fast transient simulations — a fast-simulation trades-off against the precision of the simulated spike-timing.

Finally, the simulator also allows us to program and simulate noise and mismatch effects in the neuron parameters, such as mismatch in the spiking threshold and time-constants. The Spiker simulator was used to run all the SNN simulations in this paper.

## F    Visualizing mapping of a recurrent SNN

The top row of Figure F.3 shows the mapping mechanism in action for a single test case. We note that as the depth of the network increases, the quality of fit degrades, but is still close to perfect as measured by the NMSE metric. The bottom row of Figure F.3 shows the mapping mechanism in action for devices with a very large $c_{v_p} = 1$. We note that even in such cases, the performance in lower layers remains fairly stable and only gradually degrades as the depth increases.

## G    Convergence plots for the Google spoken commands task

Figure G.4 shows the convergence plots for various values of $\alpha$.

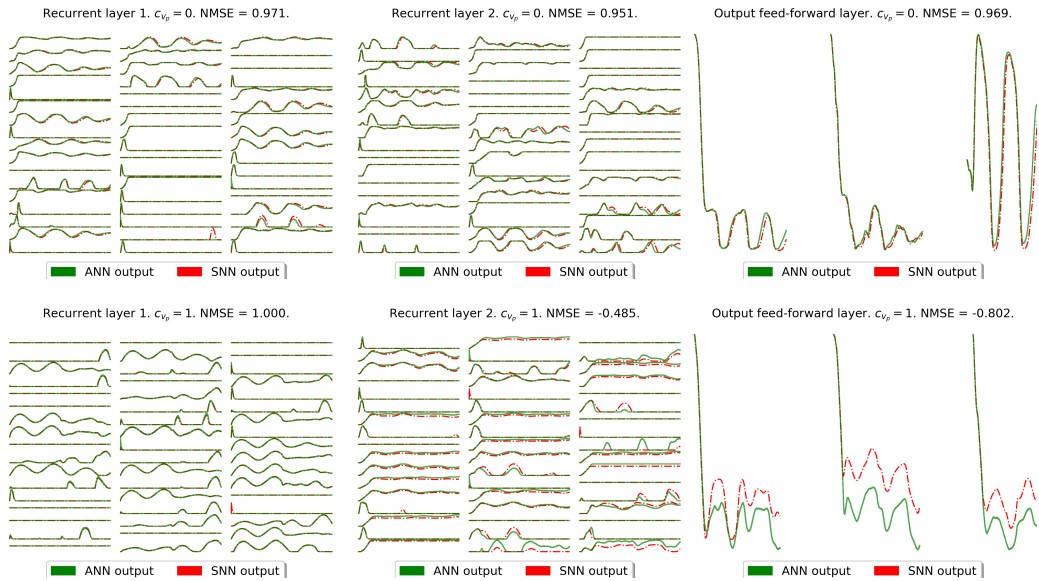

Figure F.3: Dynamics in a two-layer RNN with 51 units per layer. The SNN output shown in the figures is the trace obtain by filtering the spike trains. The NMSE measures the quality of fit with 1 indicating a perfect match and $-\infty$ a very bad fit as described in Equation 7.

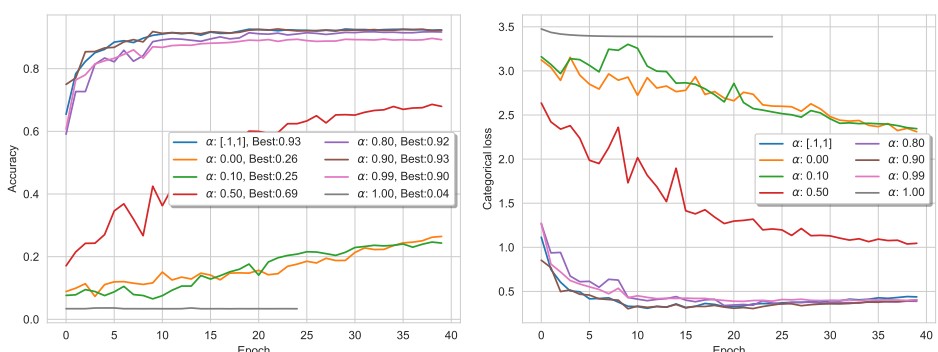

Figure G.4: Performance of the lpRNN cell on the Google commands detection task: Categorical accuracy (Left) and Cross entropy loss (Right). We note that as higher values of $\alpha$ and random sampling results in faster convergence and better accuracy.

## H MAPPING THE GOOGLE SPOKEN COMMANDS NETWORK

Each 1-second recording from the dataset is transformed using the Mel-spectrogram into 25 frequency and 128 temporal bins. This sequence represents a single speech command for the ANN. Mapping the trained ANN to an SNN is challenging because of the limitation described earlier; A short length sequence does not give the SNN enough time to generate the spikes required to transmit information. On the other hand, it is too difficult to train a longer length sequence that is several thousand samples long. To address this issue, we train the ANN using the short sequence with 128 bins and demonstrate the mapped SNN by feeding it the same spectrogram data that is up-sampled to 1 MHz. The quality of the mapped recurrent SNN is demonstrated using the weights from the trained recurrent ANN for a single case in Figure H.5. The bottom row of Figure H.5 also shows the mapping mechanism in action for RNN layers that are affected by mismatch with $c_{v_p} = 0.2$. The results indicate an excellent fit between the mapped and the original networks, even with mismatch. The prediction made by the SNN also matched that of the ANN. Unfortunately, it is computationally prohibitive to compare the

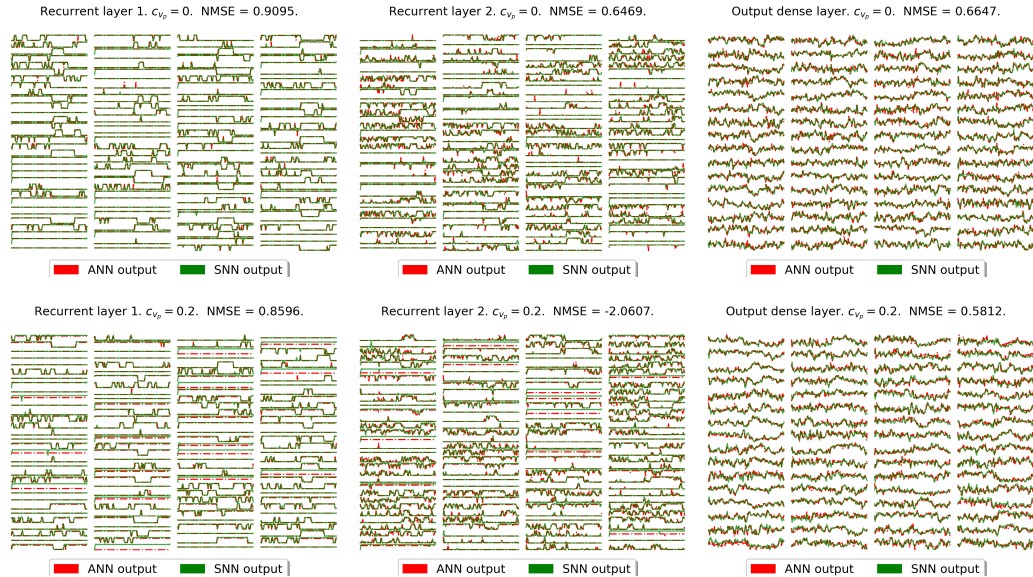

Figure H.5: Mapping a 2-layer 128 unit recurrent ANN trained to discriminate commands from the Google speech dataset to an equivalent recurrent SNN. The top row shows the mapping in action for neuron models unaffected by mismatch and the bottom row demonstrates it for mismatched units. The lpRNN cells have $\alpha = 0.99$.

accuracy results of the SNN to the reference model on the full dataset. We only demonstrate the mapping accuracy for the two recurrent layers of the network in Figure H.5 for a single command in this paper. A complete analysis of the accuracy performance will require testing on a neuromorphic system and will be the focus of a follow-up work.

# I    EXTENDING THE LOW-PASS FILTERING IDEA TO OTHER RNN MODELS

Our goal with introducing lpRNN cell was to enable faster and better training of SNNs. However, the analysis performed here indicates that low pass filtering also provides temporal regularization features which can benefit ANN-RNNs such as LSTMs. Therefore, we propose to extend the LSTM formulation by applying a low pass filter at the output ($h$), and call it an **lpLSTM** cell:

$$\text{Forget gate: } f_t = Sigmoid(W_f x_t + W_{rec_f} h_{t-1} + b_f)$$
$$\text{Input gate: } i_t = Sigmoid(W_i x_t + W_{rec_i} h_{t-1} + b_i)$$
$$\text{Output gate: } o_t = Sigmoid(W_o x_t + W_{rec_o} h_{t-1} + b_o)$$
$$\text{State}: c_t = f_t \odot c_{t-1} + i_t \odot Relu(W_c x_t + W_{rec_c} h_{t-1} + b_c)$$
$$\text{Output}: \bar{h}_t = o_t \odot Relu(c_t)$$
$$\text{Filtered Output}: h_t = \alpha \odot h_{t-1} + (1 - \alpha) \odot \bar{h}_t \tag{24}$$

where, $W_{rec_x}$, $W_x$, $b_x$ indicate the recurrent kernel, input kernel, and bias for the corresponding gate or state. Similar formulations for other RNN cells such as GRU (Chung et al., 2014), IndRNN (Li et al., 2018), Phased-LSTMs (Neil et al., 2016), Convolutional LSTMs (Xingjian et al., 2015), etc can be easily made.

## I.1    EXPERIMENTAL RESULTS FOR THE LPLSTM CELL

In this section, we benchmark the Low-Pass Long Short-Term Memory (lpLSTM) cells to their unfiltered variants. In our experiments, the learning rate was set to 0.005 and normalized gradient was clipped to 1. Current works describe use of various task-specific initialization constraints to solve the addition and copying tasks better (Henaff et al., 2016; Le et al., 2015). Instead of that, we use a

data-driven **curriculum learning protocol** in our experiments and are able to obtain dramatically improved performance on these tasks. The networks used for the copying and addition tasks are fairly small. We also test it on a character-level language modelling task with the Penn Treebank dataset (Marcus et al., 1994) using a large network with roughly 19M parameters(Kim et al., 2016). We swap the lpLSTM cells with an LSTM cells in our experiments and the network architecture and hyper-parameter settings were left unchanged from the values reported by the original authors. A summary of our observations are as follows: The lpRNN cell exhibits a dramatically improved performance over the SimpleRNN cell. The low pass filter appears to have a temporal stabilization effect even for LSTM cells. However, when other regularization and stabilization techniques such as Dropout(Srivastava et al., 2014) are introduced, the benefit appears muted. It is possible that the large networks with low-pass RNN layers require network architecture tweaks to benefit from the filtering property, but it was not investigated in this work.

### I.1.1 TEMPORAL ADDITION TASK

The addition task (Le et al., 2015) involves processing two parallel input data streams of equal length. The first stream comprises random numbers $\in (0, 1)$ and the second is full of zeros except at 2 time steps. At the end of the sequence, the network should output the sum of the two numbers in the first data stream corresponding to the time-steps when the second stream had non-zero entries. The baseline to beat is a mean square error (mse) of 0.1767, corresponding to a network that always generates 1.

We first train the RNN cell being tested on a short sequence and progressively increase the length. Each curriculum used 10,000 training and 1000 test samples. The results are shown in Figures I.6, where each stage of the curriculum learning process is marked with bands of different colours. The width of the band indicates the number of epochs taken for convergence. The length of the task was incremented when the mse went below 0.001. With random initialization, the SimpleRNN cell failed to converge beyond sequence length 40, even with curriculum learning. On the other hand, both the lpRNN and LSTM cells benefit from the curriculum learning protocol. The lpRNN cell was able to transfer learning for sequences shorter than 150 steps. While, the benefits of curriculum learning appears to have reduced beyond that, the lpRNN cells were able to achieve better than 0.001 mse for sequences up to 642. The performance of the lpRNN cell is at par or slightly inferior to other works in literature (Hochreiter and Schmidhuber, 1997; Arjovsky et al., 2016; Le et al., 2015; Hu et al., 2018), we achieved this result purely by random initialization. Another interesting outcome of this experiment was the effectiveness of a 2-unit LSTM cell in solving this task. It was was able to add sequences much longer(we tested up to 100K) than any reported work (where the networks are only able to solve the task for about 1/100th of the sequence length).

Given the effectiveness of the training protocol, we made the task more complex by allowing the second stream to have up to 10 unmasked entries during training. The trained cell was tested with a data stream having more than 10 masked entries. The LSTM cell was successful in solving this problem a mse less than 1e-3, indicating that it learnt a general add and accumulate operation. Figure I.6c shows the evolution of the gating functions, internal state, and the state variables of an LSTM cell that was trained only on a fixed length sequence of 100 with mse less than 0.001. Contrast this against the stable dynamics of the network trained by curriculum learning in Figure I.6d in a 100K sequence with mse less than 1e-3 (Figure I.6) indicating almost perfect long-term memory and addition. To our knowledge, this kind of generalized learning by an LSTM cell has not been shown before.

### I.1.2 TEMPORAL COPYING TASK

We train the RNN cells on a varying length copying task as defined in (Graves et al., 2014) instead of the original definition (Hochreiter and Schmidhuber, 1997; Arjovsky et al., 2016). This problem is harder to solve than the temporal addition task. The network receives a sequence of up to S symbols (in the original definition, S is fixed) drawn from an alphabet of size K. At the end of S symbols, a sequence of T blank symbols ending with a trigger symbol is passed. The trigger symbol indicates that the network should reproduce the first S symbols in the same order. We first train the network on a short sequence (T=3) and gradually increase it (T≤200). The sequence length is incremented when the categorical accuracy is better than 99%.

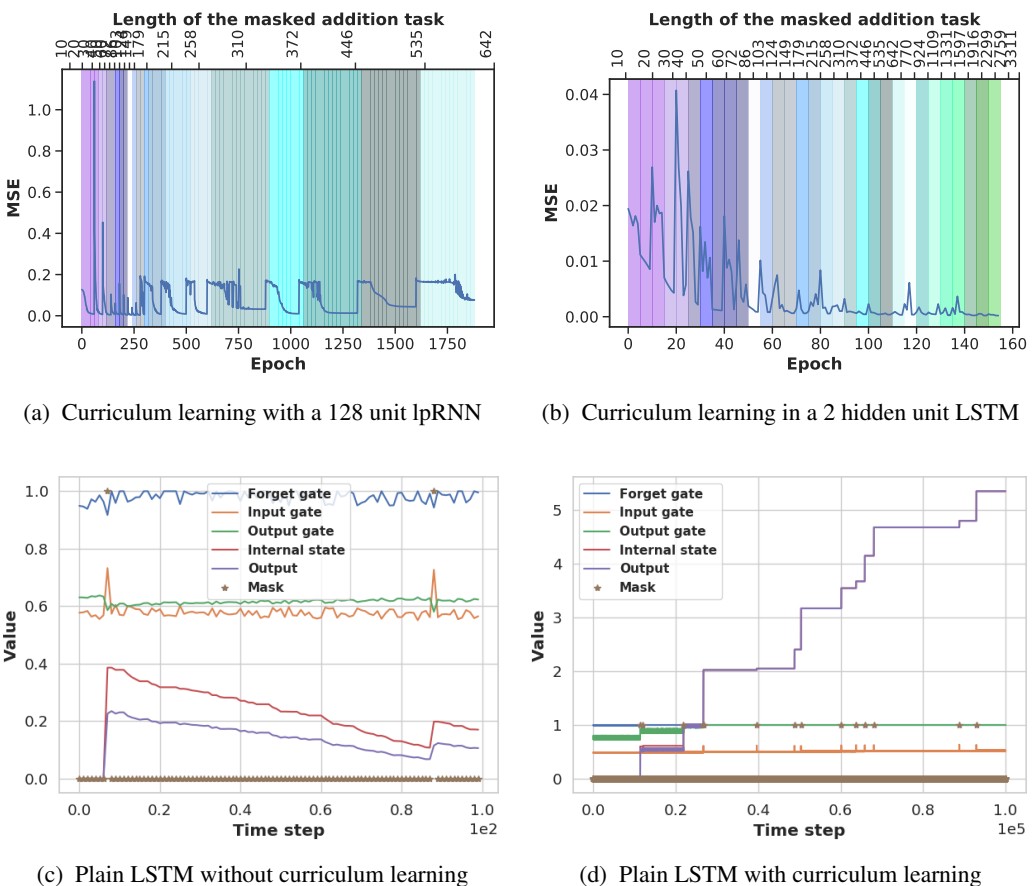

(a) Curriculum learning with a 128 unit lpRNN

(b) Curriculum learning in a 2 hidden unit LSTM

(c) Plain LSTM without curriculum learning

(d) Plain LSTM with curriculum learning

Figure I.6: Curriculum learning on the masked addition task. LSTM cell trained without curriculum learning results in unstable state variables (c). When trained with curriculum learning it looks much more stable (d). Stars in (c) and (d) indicate value of the add mask.

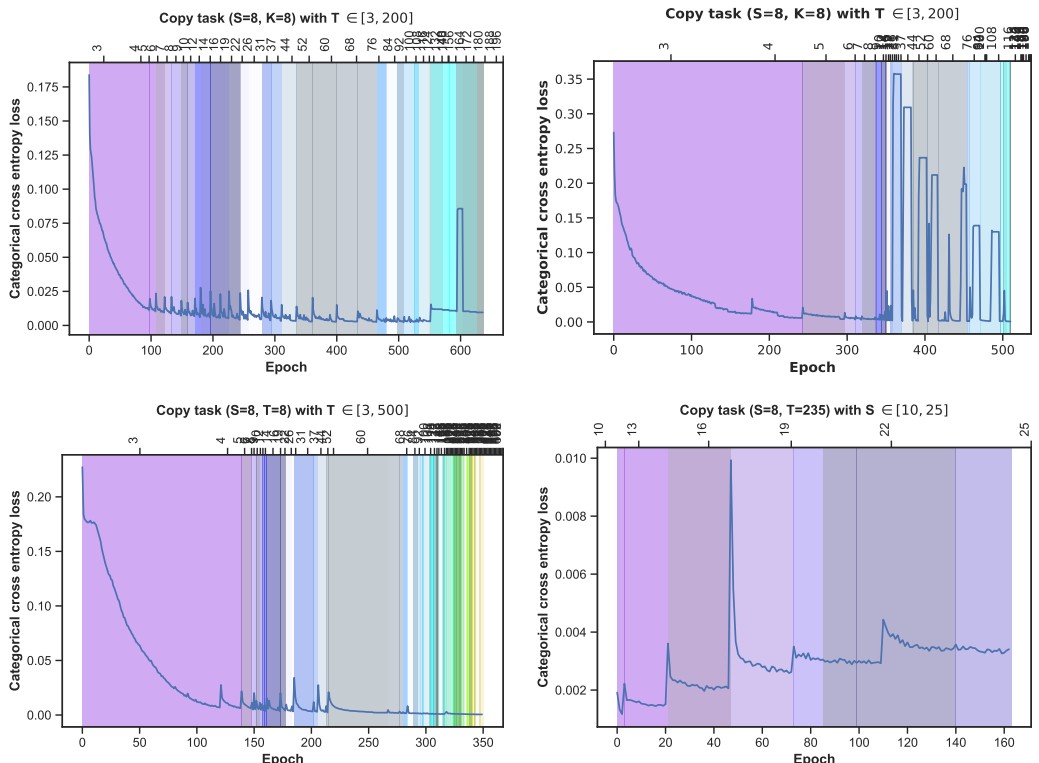

Figure I.7: Curriculum learning on the variable length copying task for a 256 unit lpRNN (top left) and a 128 unit LSTM cell (top right) and a 128 unit lpLSTM cell(bottom left and right).

The SimpleRNN cell failed at this task even for T=30 with categorical accuracy dropping to 84% when it predicted only S + T blank symbols. The lpRNN cell was able to achieve 99% accuracy for up to 120 time steps. After that, it generates T blank entries accurately but the accuracy of the last S symbols drops (For T=200, it was 96%). However, the LSTM cell achieves more than 99.5+% accuracy for all tested sequence lengths, highlighting the advantage of curriculum learning. This is a big improvement over reported results (Arjovsky et al., 2016; Graves et al., 2014; Bai et al., 2018) where LSTM cells solved the task for much smaller values of $S$ and $T$.

We observed stability issues when training an LSTM cell for sequences longer than 30, even if it eventually converged by using smaller learning rates and gradient norm scaling. This makes a good test case to validate the temporal regularization property of the lpLSTM cell. In our tests, the lpLSTM cell converged without instability with categorical accuracy higher than 99.5% for all tested values of T($\in [3, 500]$). It also to generalized larger values of S than the other cells ($\leq 25$). The lpLSTM cell exhibited a gradual degradation in performance for larger values of S. We stop our simulations when the categorical accuracy fell below 96%. These results are summarized in Figure I.7.

### I.1.3 PENN TREEBANK (PTB) CHARACTER MODEL

We studied temporal regularization in a network trained on the PTB dataset (Marcus et al., 1994) by replacing the LSTM cells by its low pass variants. We choose a model with 19M parameters (Kim et al., 2016) and trained all variants using the same settings as described in (Kim et al., 2016) for 25 epochs. We note that both lpLSTM cells converge to a better score on the training set and a marginally poorer score on the train/validation set (refer Table 6). The lpLSTM cell with relu activation also converges unlike the plain relu LSTM cell validating our claim on temporal regularization.

Table 6: Impact of temporal regularization on the Penn Treebank model.

|  | **Activation** | **Train perplexity** | **Validation Perplexity** | **Test Perplexity** |
|---|---|---|---|---|
| LSTM | relu | approx. 641 | approx. 641 | **Fails to converge** |
|  | tanh | 46.0948 | 83.9807 | 80.0873 |
| lpLSTM | tanh | 41.0545 | 84.6127 | 81.7519 |
|  | relu | 43.0602 | 84.1484 | 80.6946 |

## J    ENERGY CONSUMPTION ESTIMATION

In this section, we describe the procedure used for comparing the power consumption of the in-memory architecture against a Cortex-M4 processor. This processor was chosen as it is one of the most common low-power MCU platforms in use today.

In our analysis, we make a highly-optimistic estimate for the performance of the Cortex-M4 (ARM, 2019). We assume that there are no cache misses, that multiply and add operations take one clock cycle, the read from DRAM only consumes 6 pJ/bit, and also assume that the MCU is fully available for RNN computation. In particular, note that the memory cost per DRAM access should also include the address and data bus power consumption. This has been completely ignored in this analysis to keep things highly optimal on the Cortex-M4 side. We see that in such a configuration, the Cortex-M4 consumes only a few mW of power. In practice, the active power consumption of such processors tend to be in hundreds of mW (Rethinagiri et al., 2014).

We estimate the performance of the in-memory unit (also in 180nm technology for the known neuron implementation (Nair and Indiveri, 2019)). These results are then tabulated and system activity is modelled for two RNN models in Figure J.8a (2 layer RNN with 128 units/layer) and Figure J.8b (4 layer RNN with 500 units/layer).

The energy cost for the Cortex-M4 is modelled by the following equation:

$$E_{tot} = (Clk_{mul} \cdot N_{mul} + Clk_{add} \cdot N_{add}) \cdot E_{clk} + M \cdot E_{mem} \tag{25}$$

where

- $E_{tot}$: Total power consumed
- $Clk_{mul}$: Number of clocks for multiply
- $N_{mul}$: Number of multiply operations in the task.
- $Clk_{add}$: Number of clocks for addition.
- $N_{add}$: Number of add operations in the task.
- $E_{clk}$: Energy consumed by the processor per clock.
- $M$: Number of memory bit accesses
- $E_{mem}$: Energy cost of a accessing a single bit.

The energy cost for the in-memory architecture is modelled by the following equation:

$$E_{tot} = (N \cdot E_{spike} + M) \cdot N_{spikes} \tag{26}$$

where

- N: Number of neurons
- $E_{spike}$: Energy per spike
- $N_{spikes}$: Total number of spikes. This is computed for the $\Sigma\Delta$ model by computing the average firing rate as a function of the desired bit precision. This is approximating by equating the desired firing rate of the $\Sigma\Delta$ neuron to that of an oversampled clock necessary to achieve a desired Signal to Noise Ratio (SNR) (Pavan et al., 2017).
- M: Memory access cost. This is modelled by the the product of the read current while a spike is active.

Each of the terms in the computation of the power consumption of the Cortex and in-memory systems are in turn calculated based on a number of hardware and operational assumptions that are listed in the tables. We note an improved energy-efficiency of several hundred times across the board in both configurations. We also note that the energy-savings is much higher for the larger network. This is simply because the number of sequential compute operations increases quadratically.

| Data, network properties and Hardware assumptions (@180 nm ULL) | Input dimension | Sampling rate | Hidden layer size | Number of hidden layers | Output size | 1-bit SRAM area, 50% fill factor ( in um2) | 1. No cache misses in Cortex. 2. Memory cost only includes DRAM access and excludes data bus. | | |
|---|---|---|---|---|---|---|---|---|---|
| | 2 | 100.0 | 128 | 2 | 36 | 4.75E-01 | | | |
| | DRAM: Energy/bit * 32 bits | ARM Cortex (Energy/MHz) | ARM Cortex (Energy/clock) | Multiply clocks | Addition clocks | Energy per spike | Spike width | Per neuron leakage + wasted | Desired SDR |
| | 1.92E-10 | 1.51E-04 | 1.51E-10 | 1 | 1 | 1.00E-11 | 1.00E-06 | 1.00E-09 | 5.00E+01 |
| **Cortex** | | Computation cost per sample | | | | | | | Total clocks needed |
| | Computational cost | Clocks needed for multiply (Input stage) | Clocks needed for add (input stage) | Clocks needed for multiply (hidden) | Clocks needed for add (hidden) | Clocks needed for multiply (output) | Clocks needed for add (output) | Total clocks per sample | |
| | | 2.56E+02 | 1.28E+02 | 3.28E+04 | 3.28E+04 | 4.61E+03 | 4.61E+03 | 7.51E+04 | 7.51E+06 |
| | Memory cost | Number of memory access (per sample, input) | Number of memory access (per sample, hiddens) | Number of memory access (per sample, hiddens) | Total number of memory accesses (read + write) | Energy cost of memory accesses in mW | Total power (excluding memory) in mW | Total power including memory (in mW) | Chip area, excluding DRAM (in mm2) |
| | | 3.84E+02 | 3.30E+04 | 4.64E+03 | 7.61E+06 | 1.46E+00 | 1.13E+00 | 2.60E+00 | 0.44 |
| **In-memory computation** | Oversampling rate | Average firing rate | Total number of neurons | Total number of spikes | Biased current amplitude per read | Memory read cost | Neuron power consumption | Total power (in mW) | Chip area (in mm2) |
| | 2.93E+01 | 1.46E+03 | 294 | 4.31E+05 | 3.20E-07 | 1.38E-18 | 4.60E-06 | 4.60E-03 | 7.19E-01 |
| **Factor of improvement in energy-efficiency (compute +memory)** | 564.40 | | | | | | | | |
| **Factor of improvement in energy-efficiency (free compute, memory access pattern ideal)** | 317.71 | | | | | | | | |

(a) For a two-layer RNN with 128 units per layer

| Data, network properties and Hardware assumptions (@180 nm ULL) | Input dimension | Sampling rate | Hidden layer size | Number of hidden layers | Output size | 1-bit SRAM area, 50% fill factor ( in um2) | 1. No cache misses in Cortex. 2. Memory cost only includes DRAM access and excludes data bus. | | |
|---|---|---|---|---|---|---|---|---|---|
| | 2 | 100.0 | 500 | 4 | 36 | 4.75E-01 | | | |
| | DRAM: Energy/bit * 32 bits | ARM Cortex (Energy/MHz) | ARM Cortex (Energy/clock) | Multiply clocks | Addition clocks | Energy per spike | Spike width | Per neuron leakage + wasted | Desired SDR |
| | 1.92E-10 | 1.51E-04 | 1.51E-10 | 1 | 1 | 1.00E-11 | 1.00E-06 | 1.00E-09 | 5.00E+01 |
| **Cortex** | | Computation cost per sample | | | | | | | Total clocks needed |
| | Computational cost | Clocks needed for multiply (Input stage) | Clocks needed for add (input stage) | Clocks needed for multiply (hidden) | Clocks needed for add (hidden) | Clocks needed for multiply (output) | Clocks needed for add (output) | Total clocks per sample | |
| | | 1.00E+03 | 5.00E+02 | 1.00E+06 | 1.00E+06 | 1.80E+04 | 1.80E+04 | 2.04E+06 | 2.04E+08 |
| | Memory cost | Number of memory access (per sample, input) | Number of memory access (per sample, hiddens) | Number of memory access (per sample, hiddens) | Total number of memory accesses (read + write) | Energy cost of memory accesses in mW | Total power (excluding memory) in mW | Total power including memory (in mW) | Chip area, excluding DRAM (in mm2) |
| | | 1.50E+03 | 1.00E+06 | 1.80E+04 | 2.04E+08 | 3.92E+01 | 3.08E+01 | 7.00E+01 | 0.44 |
| **In-memory computation** | Oversampling rate | Average firing rate | Total number of neurons | Total number of spikes | Biased current amplitude per read | Memory read cost | Neuron power consumption | Total power (in mW) | Chip area (in mm2) |
| | 2.93E+01 | 1.46E+03 | 2,038 | 2.98E+06 | 3.20E-07 | 9.55E-18 | 3.19E-05 | 3.19E-02 | 1.65E+01 |
| **Factor of improvement in energy-efficiency (compute +memory)** | 2,195.46 | | | | | | | | |
| **Factor of improvement in energy-efficiency (free compute, memory access pattern ideal)** | 1,230.42 | | | | | | | | |

(b) For a four-layer RNN with 500 units per layer

Figure J.8: Energy consumption comparison between an in-memory RNN accelerator implementing the lpRNN model against a Cortex M4.

