# OpenReview forum: "High performance RNNs with spiking neurons"
_ICLR.cc/2020/Conference — Reject_

### Official Review · AnonReviewer1 · 2019-10-23
**Official Blind Review #1**

**Rating:** 6

**Review:**

This paper deals with neuromorphic computing architectures for solving deep learning problems. Given that the energy-efficiency is a critical need for modern machine learning tools, this is an important study in my opinion. Until now, most of the spiking neuron based works don't really apply for realistic deep model learning. Therefore, advancements such as these would be very useful.

However, my biggest concern is that the paper may be a bit difficult to grasp for the general ML audience. I myself am not an expert in this topic and had quite a bit of difficulty as an ML researcher in understanding parts of the paper. As a matter of fact, some of the supplementary material was quite helpful to understand the entire context. So, I suggest to move some of the basics from there to the main paper. Also, some of the performance comparison results in the supplementary section are more convincing and clear compared to the results in the main paper.

Based on my understanding, the algorithmic approach seems logical and the empirical results are convincing. However, I am not in a position to assess the level of novelty. So, I am giving the benefit of the doubt to the authors, given that this is a critical topic for the ML community.

**Experience Assessment:**

I do not know much about this area.

**Review Assessment: Checking Correctness Of Derivations And Theory:**

I assessed the sensibility of the derivations and theory.

**Review Assessment: Checking Correctness Of Experiments:**

I assessed the sensibility of the experiments.

**Review Assessment: Thoroughness In Paper Reading:**

I read the paper at least twice and used my best judgement in assessing the paper.

---

> ### Author Response · Authors · 2019-11-08
> **Added new motivation section, energy-efficiency metrics, and extended conclusion**
>
> Dear Reviewer
> Thank you for your review. We agree that energy is a key consideration for progress in ML and is it is the main motivation for our research in neuromorphics. However, the topic is challenging (and exciting) because of the little overlap between traditional chip design and ML skillsets. We think there are many interesting breakthroughs to be had at this intersection. Therefore, our goal is to make this paper intellectually accessible for the general ML audience and neuromorphic chip designers.
>
> Your comment that the paper could be too difficult to grasp for a general ML practitioner was well-received and we have tried to address this in the following ways:
> 1. Added a new motivation section that describes why neuromorphic chips are more efficient and elaborated on the theoretical argument for why that is the case.
> 2. We have also added a fairly detailed performance estimation in terms of energy cost to derive how much efficient such a framework would be in comparison to a popular mW-range edge MCU.
> 3. We have simplified the text across the paper to make it easier to follow without over-simplifying the message.
> 4. We have added a new table to complete the empirical analysis of the mapping algorithm. A second new table was added to summarize the performance of the lpRNN cell on key benchmarks. We could not find enough space in the paper to move more content from the experimental results in the supplementary section. It would be very useful if you could highlight exactly what was unclear and we would be happy to make the necessary revision.
> 5. We have tried to make the novelty clearer with explicit claims about what makes this work important through the paper and in the conclusion. In summary, the novelty lies in the signal processing viewpoint to modeling spiking (and non-spiking) neurons and the resulting mapping mechanism. We think that the approach to obtaining the model and the mapping technique are important ideas that will benefit this field significantly.
>
> Thank you very much for your time and we hope this response and the revision addresses your concerns.

---

### Official Review · AnonReviewer3 · 2019-10-24
**Official Blind Review #3**

**Rating:** 6

**Review:**

The paper promises efficient training of spiking neuron models using back-propagation. The authors say that this is important because spiking networks offer significant energy savings, yet they typically perform poorly compared to prevalent artificial neural networks (ANNs). They aim to support this with several synthetic and one real-world task.

I selected to reject the paper based on the support for the main claim of the paper being weak and inadequately demonstrated. No comparisons are to baselines are presented in the text. Reproducibility is hampered by a great deal of the experimental details missing.

On the experimental results:
* In section 4.1, some justification is needed for why these experiments are relevant.
* In sections 4.4 and 4.5, experiments are discussed, but results are not presented.
* If space is a limitation, I would strongly prefer a table of actual results for sections 4.4 and 4.5 rather than Figure 2.
* Since the encoding of the task in the spiking and non-spiking cases is so drastically different, making that crystal clear would be great.

Other comments:
* The goal of energy savings is very appealing, so it would help to give *some* quantitative measure of savings promised.
* Similarly, the claim about the poor performance of spiking neurons would benefit from some quantitative measures.
* A figure showing the operation of the novel SigmaDelta neuron, compared to the standard aI&F neuron, would be welcome.

Overall I think the paper would benefit from focussing much more on the results, making the claims as quantitatively apparent as possible.



**Experience Assessment:**

I do not know much about this area.

**Review Assessment: Checking Correctness Of Derivations And Theory:**

I assessed the sensibility of the derivations and theory.

**Review Assessment: Checking Correctness Of Experiments:**

I assessed the sensibility of the experiments.

**Review Assessment: Thoroughness In Paper Reading:**

I read the paper at least twice and used my best judgement in assessing the paper.

---

> ### Author Response · Authors · 2019-11-08
> **Paper revision to address your concerns**
>
> Dear Reviewer
> Thank you very much for the clear and specific criticism of our paper. We found all of them constructive and have addressed each of them in the revision. We have in particular added clear measures of the expected energy benefits.
>
> 1. "main claim of the paper being weak and inadequately demonstrated":
> Both the approach and algorithm presented in this paper represent important  breakthroughs for the field of neuromorphics. We claim and prove that
> a. A neuron is an LPF: This is known in neuroscience. We show theoretically that a Sigma-Delta neuron is aN LPF.
> b. The low pass RNN can be mapped to spikes: We show this extensively in Tables 1-3, using a custom transient simulator for a full-chip model. We also provide theory for why it works and its limitations, both of which are demonstrated empirically. Such a demonstration has not been made before, to our knowledge.
> c. The lpRNN achieves significantly better performance on relevant benchmarks: We demonstrate this on two synthetic and one audio task. We do not present results for tasks that require larger networks in this paper, as neuromorphic chips tend to have limited neuron resources.
> d. This model is energy-efficient. This is an addition in the revision. Based on a very conservative model, we expected at least a 500x reduction in energy cost vs a popular embedded MCU.
>
> * We have also made several improvements to address your specific concerns.*
>
> 2. "In section 4.1, some justification is needed for why these experiments are relevant."
> SDR in the spiking neuron is analogous to low bit-precision in conventional ANNs. It is important to measure how much precision is available using a metric that is meaningful for the proposed spiking architecture. We have now added this to the revised text.
>
> 3. " In sections 4.4 and 4.5, experiments are discussed, but results are not presented."
> We have added a table tabulating the final performance metrics. Space restrictions limit us from including the convergence plots and some other insights, which are included in the supplementary material.
>
> 4. * If space is a limitation, I would strongly prefer a table of actual results for sections 4.4 and 4.5 rather than Figure 2.
> We think that Figure 2 is an important one to establish the performance of the neuron model, especially for designers of neuromorphic systems. There is a clear need to identify standard metrics for performance and plots like these are typically very insightful for chip designers.
>
> 5. * Since the encoding of the task in the spiking and non-spiking cases is so drastically different, making that crystal clear would be great.
> We have simplified the text in the paper to make this clearer, but it is possible that we may not have addressed your concern exactly. We are happy to make revisions to make this clearer.
>
> 6. * The goal of energy savings is very appealing, so it would help to give *some* quantitative measure of savings promised.
> Thank you very much for this comment. We have now added a full section and an extensive analysis to estimate the energy efficiency achievable with this system. In fact, it is now a key highlight of the paper.
>
> 7. * Similarly, the claim about the poor performance of spiking neurons would benefit from some quantitative measures.
> This is difficult because there are no spiking neural network implementations that solve the sophisticated tasks we discuss in the paper. There has been some work in mapping RNNs to a spiking framework [1, 2 (published days ago)]) using rate coding on much simpler networks and tasks. Rate coding is highly inefficient as the resultant high firing rates cost a lot of energy [3]. For example, 8-bit precision requires O(256) spikes per sample! Moreover, these models/techniques also do not take advantage of the low-pass filtering property of a neuron. Finally, the performance of the mapping technique is difficult to benchmark against as there are no quantitative metrics in published literature. We believe our paper is an important step in addressing these shortcomings. We understand this is a strong claim, but this is true to the best of our knowledge. We are of course happy to be corrected.
>
> 8. * A figure showing the operation of the novel SigmaDelta neuron, compared to the standard aI&F neuron, would be welcome.
> We have added a new Figure 1b to highlight this clearly.
>
> Thank you very much for your time, consideration, and thoughtful comments.
>
> [1] Diehl, Peter U., et al. "Conversion of artificial recurrent neural networks to spiking neural networks for low-power neuromorphic hardware." 2016 IEEE International Conference on Rebooting Computing (ICRC). IEEE, 2016.
> [2] Kim, Robert, Yinghao Li, and Terrence J. Sejnowski. "Simple Framework for Constructing Functional Spiking Recurrent Neural Networks." bioRxiv (2019): 579706.
> [3] Nair, Manu V., and Giacomo Indiveri. "An ultra-low power sigma-delta neuron circuit." 2019 IEEE International Symposium on Circuits and Systems (ISCAS). IEEE, 2019.

---

### Official Review · AnonReviewer2 · 2019-10-24
**Official Blind Review #2**

**Rating:** 1

**Review:**

The submission describes an adaptive spiking neuron model that is based on the Laplace transform of the model output.
This reformulation allows to train a recurrent neural network of spiking neural network with good performances.
Three tasks are proposed to assess the recurrent net, on synthetic data and real signal.

I think this contribution could not be accepted for a methodological problem: the main idea is to use neuromorphic chips that use spiking neural networks with an approximation that is basically similar to an artificial neural network. The authors need to introduce a complex rescaling of the time step, repercussion on the computation. This point should be shown experimentally. The authors introduced a modification of the aI&F and are applying low pass filtering the spike trains.

As all the results are obtained on a python simulator, it is difficult to assess the interest and the applicability to real neuromorphic chips.
As pointed out by the authors, the main interest of these architectures are the lower energy consumption than CPU/GPU-based architecture.
Unfortunately, it is not possible to assess if the proposed neuron model is working on energy efficient architecture.


[1] Nair, M. V. and Indiveri, G. (2019). An ultra-low power sigma-delta neuron circuit. In 2019 IEEE International Symposium on Circuits and Systems (ISCAS), pages 1–5.


**Experience Assessment:**

I have published one or two papers in this area.

**Review Assessment: Checking Correctness Of Derivations And Theory:**

N/A

**Review Assessment: Checking Correctness Of Experiments:**

I assessed the sensibility of the experiments.

**Review Assessment: Thoroughness In Paper Reading:**

I read the paper at least twice and used my best judgement in assessing the paper.

---

> ### Author Response · Authors · 2019-11-05
> **Methodology**
>
> Dear Reviewer
> Thank you for reviewing our work. In this paper, we present a new mechanism for training models suitable for neuromorphic chips. To our knowledge, the complexity of tasks and the extensive demonstration of mapping using a non-rate-based model is unprecedented for SNNs. We think that this work is an important breakthrough both in terms of the techniques we introduce and the inter-disciplinary approach to neuromorphics. We argue that we have provided very strong evidence for all these points and will attempt to convince you of this with the following arguments:
>
> 1. "The authors need to introduce a complex rescaling of the time step, repercussion on the computation. This point should be shown experimentally."
> Answer: We have made this demonstration the core idea of the paper, both theoretically and empirically. The results in Table 1 and 2 are the performance of the rescaling technique for over 100 sample networks per data point for deep and wide networks. We also show in Table 5 (in the revised draft) that the mapping is accurate in the sub-sampling case. If you disagree, could you please clarify why with more specifics?
>
> 2. "As all the results are obtained on a python simulator, it is difficult to assess the interest and the applicability to real neuromorphic chips."
> Answer: The purpose of this paper is to present an algorithm that is compatible with known neuromorphic chips. Demonstration on a real chip is outside its scope, but we do provide comprehensive evidence for the model’s applicability to such chips. There are three aspects to consider to answer this question, and we provide evidence for all three:
> a. The neuron model
> b. The simulation of the complex network
> c.The possibility to implement the network on a chip.
>
> a. The sigma-delta loop is well studied and its equivalence to a filter is well-established in literature and practice [5]. There is literature that proves that the specific neuron model is functional [6, 7].  Moreover, a CMOS mixed-signal circuit implementation for the neuron model has been published in the paper that the reviewer cited [1].
>
> b. Implementing several hundred neurons on a chip is a complex and expensive task. Such a system is always designed by first simulating them on transient circuit simulators based on SPICE or similar. The Python simulator is exactly that. This is standard practice in any mixed-signal circuit design in the industry and academia. We created a new simulator to allow ease of integration to the Python-based NN frameworks! This must be seen as a contribution to the work and not as a drawback. The code for this simulator has now been released.
>
> c. The ability to integrate a mixed-signal neuron unit in a neuromorphic chip has been demonstrated extensively in the literature[2, 3, 4]. The exact system architecture described in Fig D.2 has also been used in [2]. The argument in this paper is that if we replace the neuron circuit in [2] with the sigma-delta neuron, train it using the procedure described in the paper, we will achieve high accuracy performance and simple training. We think that this is an important breakthrough for the field.
>
> 3. Finally, based on reviewer inputs, we have made significant improvements to the paper. Note in particular the section on the energy-consumption estimates and added mapping results for the sub-sampling case. The near-perfect mapping of the model (Cvp > 0) is proof that the models we use for the complex tasks are mappable.
> =====
> *We hope this satisfies your concerns. Thank you for your time and consideration.*
> =====
> [1] Nair, M. V. and Indiveri, G. (2019). An ultra-low-power sigma-delta neuron circuit. In 2019 IEEE International Symposium on Circuits and Systems (ISCAS).
> [2] Qiao, N., Mostafa, H., Corradi, F., Osswald, M., Stefanini, F., Sumislawska, D., and Indiveri, G. (2015). A reconfigurable on-line learning spiking neuromorphic processor comprising 256 neurons and 128k synapses. Frontiers in neuroscience.
> [3] Moradi, S., Qiao, N., Stefanini, F., and Indiveri, G. (2018). A scalable multicore architecture with heterogeneous memory structures for dynamic neuromorphic asynchronous processors (DYNAPs). Biomedical Circuits and Systems, IEEE Transactions on.
> [4] Frenkel, C., Lefebvre, M., Legat, J.-D., and Bol, D. (2019). A 0.086-mm2 12.7-pj/sop 64k-synapse 256-neuron online-learning digital spiking neuromorphic processor in 28-nm CMOS. IEEE transactions on biomedical circuits and systems.
> [5] Pavan, S., Schreier, R., and Temes, G. C. (2017). Understanding delta-sigma data converters. John
> Wiley & Sons.
> [6] Yoon, Y. C. (2016). Lif and simplified srm neurons encode signals into spikes via a form of asynchronous pulse sigma–delta modulation. IEEE transactions on neural networks and learning systems, 28(5).
> [7] Bohte, S. M. (2012). Efficient spike-coding with multiplicative adaptation in a spike response model. In Advances in Neural Information Processing Systems.

---

### Author Response · Authors · 2019-11-06
**Source code released**

Dear Reviewers and Readers
We are happy to release our anonymized source code here -
https://anonymous.4open.science/r/d733c4d7-3e86-4142-965a-aa25d0b29bbe/

It includes all the code and data we used in all our experiments, including the Spiker simulator.

We will be uploading a revision of the paper shortly along with a response to your comments.

Thank you for your time and consideration.

---

### Author Response · Authors · 2019-11-08
**Paper revision uploaded**

Dear Reviewers
Thank you very much for your comments. We found them constructive and based on your inputs, we have uploaded a revised version of the paper. The changes have been highlighted in blue.

A summary of changes made to the paper follow:
1. We have added energy-efficiency estimates for the model and compared it to a popular edge MCU (500x better).
2. Added a new section after introduction to motivate the energy-efficiency argument further.
3. Added a transient time plot to highlight the operation of the asynchronous sigma-delta neuron. Also added a description of why it is advantageous.
4. Simplified the text describing the motivation for low-pass abstraction of the neuron model.
5. Added text to describe the motivation for experimental results in Section 5.1 (earlier Section 4.1).
6. Added new table comparing the mapping performance when ANNs are trained using sub-sampled data sequences that from 20K to 20 steps long.
7. Added comparison works to the three benchmarking tasks.
8. Added a section in supplementary material that describes the modeling of the reference in-memory accelerator and a Cortex M4.
9. Expanded the conclusion and add text throughout the paper to highlight the contributions and expected impact more clearly.
10. Finally, we have also released the source code for the paper.

Thank you very much for your time and consideration.

---

### Decision · Program_Chairs · 2019-12-19

**Decision:**

Reject

**Comment:**

This paper presents a new mechanism to train spiking neural networks that is more suitable for neuromorphic chips.
While the text is well written and the experiments provide an interesting analysis, the relevance of the proposed neuron models to the ICLR/ML community seems small at this point. My recommendation is that this paper should be submitted to a more specialised conference/workshop dedicated to hardware methods.

---

> ### Author Response · Authors · 2020-02-04
> **Relevance**
>
> Thank you for the good words about the paper.  We respect your descision, but humbly disagree with the reason for it.
>
> We believe that this paper is highly relevant to the AI/ML algorithms community today. There are a wave of companies and academics producing new neuromorphic hardware, such as Intel Loihi, IBM TrueNorth, Brainchip Adaptiva, Synthara, Syntiant, Mythic, SpinNaker and so on. The list is extremely long.
>
> These hardware do require a specialized algorithms. This problem deserves serious attention from the algorithms community. What better place to present this and discuss this than one of the top AI/ML conferences?
>
> Its relevance was assumed to be understood by the authors. If necessary, we would also have modified the paper to reflect this , but unfortunately, this opinion/concern was never raised during the review process!
>
> How can the paper be rejected for a reason that was never raised by the AC or the reviewers during the entire review process?